# Development of Biocompatible 3D-Printed Artificial Blood Vessels through Multidimensional Approaches

**DOI:** 10.3390/jfb14100497

**Published:** 2023-10-08

**Authors:** Jaewoo Choi, Eun Ji Lee, Woong Bi Jang, Sang-Mo Kwon

**Affiliations:** 1Laboratory for Vascular Medicine and Stem Cell Biology, Department of Physiology, Medical Research Institute, School of Medicine, Pusan National University, Yangsan 50612, Republic of Korea; wozh1304@naver.com (J.C.); easy2697@naver.com (E.J.L.); 2Convergence Stem Cell Research Center, Pusan National University, Yangsan 50612, Republic of Korea

**Keywords:** 3D printing, artificial blood vessel, biocompatible, vascular diseases, biofabrication, bioink

## Abstract

Within the human body, the intricate network of blood vessels plays a pivotal role in transporting nutrients and oxygen and maintaining homeostasis. Bioprinting is an innovative technology with the potential to revolutionize this field by constructing complex multicellular structures. This technique offers the advantage of depositing individual cells, growth factors, and biochemical signals, thereby facilitating the growth of functional blood vessels. Despite the challenges in fabricating vascularized constructs, bioprinting has emerged as an advance in organ engineering. The continuous evolution of bioprinting technology and biomaterial knowledge provides an avenue to overcome the hurdles associated with vascularized tissue fabrication. This article provides an overview of the biofabrication process used to create vascular and vascularized constructs. It delves into the various techniques used in vascular engineering, including extrusion-, droplet-, and laser-based bioprinting methods. Integrating these techniques offers the prospect of crafting artificial blood vessels with remarkable precision and functionality. Therefore, the potential impact of bioprinting in vascular engineering is significant. With technological advances, it holds promise in revolutionizing organ transplantation, tissue engineering, and regenerative medicine. By mimicking the natural complexity of blood vessels, bioprinting brings us one step closer to engineering organs with functional vasculature, ushering in a new era of medical advancement.

## 1. Introduction

Vascular diseases have a profound influence on global mortality rates, underscoring their status as a pressing public health concern. These ailments affect the well-being and quality of life of individuals. The list of conditions includes atherosclerosis, aneurysms, congenital vascular malformations, deep vein thrombosis (DVT), peripheral arteries, and stroke [1,2,3,4,5,6]. Given the confluence of factors, such as an aging population and lifestyle-related influences, the imperative for innovative treatment modalities has intensified, aiming to tackle the web of medical intricacies presented by these vascular challenges. From plaque accumulation within arteries to emerging congenital anomalies in blood vessels, these vascular conditions require diverse therapeutic strategies [7,8,9,10,11]. For instance, atherosclerosis, which is characterized by the accumulation of plaques within arteries, is often managed through lifestyle adjustments and pharmacological interventions [12,13,14,15]. In contrast, aneurysms require surgical procedures, such as clipping or endovascular coiling, to prevent catastrophic ruptures [16,17,18]. DVT is characterized by the presence of blood clots within the deep veins and requires anticoagulant regimens [19,20,21]; whereas, varicose veins are alleviated through a multitude of approaches, including compression stockings, sclerotherapy, and laser therapy [22,23,24]. The severity of peripheral artery disease, which disrupts blood flow to the lower extremities, may mandate changes in lifestyle, medication, or interventional measures, such as angioplasty or bypass surgery [25,26,27]. For the intricate landscape of vascular malformations encompassing congenital aberrations in blood vessels, the spectrum of treatment alternatives spans from embolization to surgical excision or laser-based interventions, dependent on factors such as the specific type and anatomical location [28,29,30,31,32].

Advancements in medical technology have paved the way for innovative approaches to managing vascular diseases. Among these breakthroughs, three-dimensional (3D) printing has emerged as a revolutionary technology with transformative potential in various industries including health care. In recent years, the medical field has witnessed a paradigm shift as 3D printing offers unprecedented opportunities for personalized and precise health care solutions. Its versatility allows for the creation of intricate 3D structures from digital models, making it a promising tool for addressing the complexities of vascular conditions. In the realm of vascular disease treatment, 3D printing has shown tremendous promise in enhancing diagnosis, surgical planning, and therapeutic interventions [33,34,35,36,37]. One of its most important applications is preoperative planning, in which 3D-printed anatomical models are generated based on patient-specific imaging data. These realistic replicas provide health care professionals with tangible representations of complex vascular structures, enabling a better understanding and visualization of the patient’s anatomy. Such detailed models enable surgeons to develop customized and optimized surgical strategies, reducing the risk of complications, and improving overall procedural outcomes [38,39,40].

Before the advent of 3D printing, the goal to fabricate artificial blood vessels involved early attempts using rudimentary materials such as rubber tubing and silk thread. However, these initial endeavors were fraught with biocompatibility issues and lacked suitability for integration into the intricate human circulatory system. In addition, biological alternatives emerged in the form of xenografts (animal blood vessels) and allografts (deceased human donor blood vessels). While offering certain advantages, such as biocompatibility, these biological transplants introduced new challenges, including the risk of infection and graft rejection. The mid-20th century saw a significant advancement in vascular graft technology with the development of synthetic materials such as Dacron (polyethylene terephthalate) and Teflon (polytetrafluoroethylene). These innovations significantly enhanced the biocompatibility and versatility of vascular grafts, laying the foundation for more effective and durable solutions.

3D-printed artificial blood vessels have vast potential; however, their realization is a multifaceted challenge, characterized by the intricate interplay of various methodologies. This encompassing complexity spans material selection and the diverse array of 3D bioprinting techniques employed in their creation. The choice of materials, referred to as bioinks in the realm of 3D bioprinting, is of paramount importance. These bioinks must meet stringent criteria, including biocompatibility, replication of the extracellular matrix (ECM) of natural blood vessels, and the facilitation of cell adhesion, proliferation, and tissue growth. The diversity of 3D bioprinting techniques, encompassing inkjet, extrusion-based, and laser-based bioprinting, further amplifies the intricacy of artificial blood vessel fabrication. Each technique possesses unique attributes and applications, adding layers of complexity to the already intricate process. Functional 3D-printed blood vessels take us beyond the realm of materials and techniques, venturing into the domain of cell seeding and functionalization. Typically, bioprinted constructs are enriched with endothelial cells, which play a pivotal role in forming the inner lining of blood vessels. This intricate choreography ensures that the final structures not only replicate the anatomical complexity but also function in harmony with the circulatory system. Therefore, this article explains the artificial blood vessel 3D printing method that is currently being studied, as well as the materials used and the pros and cons of each.

In addition to preoperative planning, 3D printing enables surgical simulation, a powerful training tool that allows surgeons and medical students to practice intricate procedures in risk-free virtual environments. This practice not only enhances technical skills but also fosters a deeper understanding of the unique challenges associated with each patient’s vascular disease, ultimately leading to improved surgical precision and safety [41,42]. In addition to surgical applications, 3D printing plays a pivotal role in developing patient-specific implants and prostheses for treating vascular diseases. Traditional off-the-shelf medical devices often lack a precise fit and fail to accommodate the individual patients’ anatomy. In contrast, 3D printing allows bespoke implants and prosthetics to be fabricated and tailored to each patient’s unique vascular needs [34,43,44]. This personalization enhances implant compatibility, biocompatibility, and long-term functionality, thereby improving the overall quality of life of patients undergoing vascular interventions. Moreover, the emerging field of 3D bioprinting holds promise for treating vascular diseases. Researchers are exploring the fabrication of bioengineered vascular constructs and functional blood vessels by using living cells and biocompatible materials [45,46,47,48,49,50]. The potential to replace or regenerate damaged blood vessels through bioprinting offers hope for a new era of regenerative medicine, in which vascular diseases can be treated at their root cause, ultimately revolutionizing patient care. The 3D printing of blood vessels presents unique challenges and requirements distinct from the production of solid organs through traditional layering methods. In the 3D printing of biological organs, most organs are constructed through a layered approach. This approach is effective for creating larger organs where the shape can be realized through a structured arrangement of differentiated cells. However, the situation of artificial blood vessel fabrication is notably differents. Blood vessels have a relatively thin and tubular structure, with a critical need for an empty central space to facilitate blood flow. This requirement mandates a departure from traditional layering techniques. Instead, it necessitates specialized methods such as core-shell extrusion or microprinting.

In this review, we explore the advantages and disadvantages of the materials and printing methodologies employed in the intricate process of fabricating artificial blood vessels using innovative 3D printing technology. In addition, we provide detailed insight into applying the methodologies underlying artificial blood vessel production. Finally, we focus on future prospects of artificial blood vessels and 3D bioprinting, shedding light on exciting possibilities and potential advancements in the realm of regenerative medicine and vascular therapies

## 2. Biocompatible Materials for 3D-Printed Vascular Structures

Initially, silicone emerged as a promising material for vascular applications due to its innate flexibility [51,52]. However, as the practice of using silicone for long-term vascular implants became prevalent, concerns arose regarding its vulnerability to infections and proclivity for blood clot formation [53]. These early experiences underscored the need for alternative materials with improved biocompatibility and long-term performance. In response to these challenges, the medical community transitioned to materials such as Dacron and expanded polytetrafluoroethylene (ePTFE), primarily due to their remarkable durability and facilitation of unimpeded blood flow [54]. Dacron, a synthetic polyester, has been a mainstay for vascular grafts since the mid-20th century, while ePTFE, known for its low friction and chemical resistance, found extensive use in vascular applications [55,56]. Hess et al. implanted an artificial blood vessel made of ePTFE into the aorta of a rat and con-firmed the formation of a new intima, promoting the possibility of its use as a blood vessel [57]. Although various studies have been conducted on these materials, they have limitations. Kester conducted a study that reduced the occurrence of side effects by inducing a reduction in blood clots using a platelet inhibitor drug together to reduce the formation of blood clots during vascular transplantation using Dacron [58]. In this study, Dacron and ePTFE suffer from limited bio-compatibility, making them less than ideal for promoting endothelial cell adhesion, and their high rigidity poses challenges for post-implantation flexibility. As the quest for superior vascular materials continues, recent advances have focused on the use of biomaterials. Polymers such as polyglycolic acid (PGA), polylactic acid (PLA), and polyethylene glycol terephthalate (PET) have attracted attention due to their cost-effectiveness and excellent biocompatibility. Hu et al. fabricated small-diameter blood vessels using polymer PET and judged their biocompatibility. It was shown that the adhesion and proliferation abilities of cells in blood vessels were improved, and the progress of endothelialization was confirmed to determine the biocompatibility of blood vessels [56]. In addition, Wang et al. fabricated small-diameter blood vessels using PGA, human-adipose-derived stem cells, and smooth muscle cells and confirmed their feasibility as blood vessels during transplantation [59]. However, despite these advantages, the search for materials that balance strength as a vascular graft and resistance to body corrosion is ongoing. Polylactic-co-glycolic acid (PLGA), a mixture of two polymers, has been studied as a potential solution. Morteza Bazgir, et al. confirmed the effect of polymers such as polycaprolactone (PCL) and PLGA on endothelialization during vascular transplantation and showed high cell proliferation and adhesion, showing improved properties for the construction of tissue engineering vascular grafts [60]. Despite its suitable properties in vivo, it still suffers from limitations associated with controlling its degradation rate. To address these limitations, researchers have turned to hybrid materials that incorporate polymers and metals. By leveraging the flexibility of polymers along with the rigidity of metals, these hybrid materials offer promising in vivo performance and adaptability to a variety of applications through tailored property tuning [61,62]. Mahdis Shayan et al. produced and transplanted small-diameter blood vessels using Nitinol. Although it has excellent thrombotic resistance, vascular endothelialization is not sufficient, so it is expected that it can be used in combination with silk to promote cell growth and be used as a material for artificial blood vessel transplantation [63]. Nevertheless, concerns persist about the potential for corrosion of metals in the human body, necessitating rigorous biocompatibility evaluation. Additionally, there is a burgeoning field exploring the use of ECM and decellularized ECM, dECM for artificial blood vessel construction [64,65]. dECMs, ECM-based materials, leverage the natural architecture of tissues and provide a scaffold for cellular adhesion and growth. C.Y. Xu, et al. confirmed the survival rate and adhesion ability of SMC through a biodegradable nanofiber structure and said that replicating the structure mimicking ECM could represent a tissue engineering scaffold for blood vessels [66]. These materials show promise in supporting tissue regeneration and have gained attention for their potential in vascular grafts. However, it’s important to note that ECM and dECM-based materials are not without limitations. One significant challenge is the variability in ECM composition from different tissue sources, which can impact the mechanical properties and biological response of the graft. Additionally, ensuring complete decellularization of dECM and avoiding residual cellular debris is essential to prevent adverse immunogenic reactions. The sourcing and preparation of ECM and dECM materials also raise logistical and regulatory challenges. Another research direction of artificial blood vessel production involves 3D printing techniques employing bioinks such as collagen, alginate, and hyaluronic acid (Figure 1). This approach boasts enhanced biocompatibility and improved cellular interactions [67,68]. In a pioneering study, Yuqing Niu and Massimiliano Galluzzi conducted extensive research on vascular scaffolds enriched with hyaluronic acid and collagen. Their meticulous investigations yielded compelling results, demonstrating a substantial enhancement in the expression of endothelial cell elongation, proliferation, and morphological development within the scaffold. Equally noteworthy, their study revealed a conspicuous absence of hemolysis and blood coagulation, underscoring the potential for these artificial blood vessels to be viable candidates for transplantation [69]. Additionally, a parallel effort by F. Ruther et al. and collaborators introduced a novel approach to creating artificial blood vessels using a composite of alginate and gelatin. A distinctive feature of this method is its adaptability, allowing the fabrication of vessels with varying diameters, extending up to an impressive 3.7 mm. Through rigorous experimentation, they definitively established the proliferation and migration of both fibroblasts and endothelial cells throughout the outer layer and inner core of these artificial vessels. This compelling evidence strongly supports the feasibility of utilizing this cutting-edge technology to engineer intricate artificial vascular structures [70]. However, it demands heightened stability and entails a complex and time-intensive manufacturing process. The historical evolution of artificial blood vessel materials reflects a relentless pursuit of materials that balance biocompatibility, durability, and flexibility. Each material choice brings its own set of advantages and limitations, underscoring the ongoing search for the ideal material for vascular grafts.

### 2.1. Exploring Biodegradable Polymer Selection for Enhanced 3D-Printed Artificial Blood Vessel

In the 3D-printed artificial blood vessels, biodegradable polymers play a pivotal role. Among these, polylactic acid stands out as an environmentally-sound choice sourced from renewable materials, such as corn starch [51,52,53,54]. However, its brittleness and accelerated degradation kinetics might restrict its application in certain tissue engineering contexts. Conversely, polycaprolactone exhibits a slower degradation rate and flexibility, rendering it a promising candidate for long-term tissue regeneration. However, its lower mechanical strength might constrain its use in scenarios that demand substantial load-bearing capacities. The PLGA, which is a combination of PGA and PLA, offers the advantage of biocompatibility, high degradability, and tunable physical properties, making it versatile in medical applications. However, the resultant acidic degradation byproducts could influence the local tissue pH, and PLGA’s inherent hydrophobicity could impede optimal cell interactions [55,56,57,58,59,60,61,62,63]. Polydioxanone is appropriated by its flexibility and gradual degradation, making it suitable for dynamic tis-sue-engineering applications. However, its high cost and inadequacy in situations requiring prompt tissue remodeling might limit its widespread use [64,65,66,67]. In the intricate process of selecting an optimal polymer, considerations (such as the intended application, mechanical attributes, degradation kinetics, biocompatibility, and cellular responses) are of paramount importance. Moreover, the artful blending of different polymers or strategic amalgamation with complementary materials has emerged as a promising strategy to offset individual polymer weaknesses and tailor material properties to the nuanced requirements of artificial blood vessel fabrication.

### 2.2. Advancing Hydrogel Suitability for 3D-Printed Vascular Structures

Alginate hydrogels, which are obtained from marine sources, exhibit suitability for 3D bioprinting owing to their rapid gelation upon exposure to calcium ions. This attribute facilitates precise deposition during the printing process. However, a drawback is that their modest mechanical strength limits their viability in scenarios requiring substantial load-bearing capacity. In addition, while being adept at fostering cell encapsulation, alginate hydrogels might not replicate the intricate biochemical and mechanical cues inherent to native tissue microenvironments [71,72,73,74,75]. Conversely, gelatin hydrogels derived from collagen (a key component of the ECM) show biocompatibility and offer an environment conducive to cell attachment and proliferation. Controlled gelation achieved through enzymatic crosslinking enhances printability, albeit at the expense of vigilant temperature control during the printing process. However, their mechanical robustness lags and vulnerability to degradation might curtail their application in contexts that demand sustained mechanical resilience [76,77,78,79,80]. Similarly, agarose hydrogels, commonly employed in 3D cell cultures, can preserve cell viability and maintain well-defined structures. This attribute makes them valuable substrates for mimicking cellular behavior and interactions in a controlled laboratory setting. However, their mechanical strength is low, rendering them unsuitable for applications requiring structural integrity and endurance [81,82,83,84]. Fibrin (a natural protein involved in blood clotting and wound healing) is a valuable hydrogel for 3D printing. It offers excellent biocompatibility and supports cell attachment and migration. Its resemblance to the ECM and its ability to incorporate growth factors make it a versatile choice. However, its rapid degradation might limit its use in long-term tissue engineering projects [85,86,87,88,89,90]. Ultimately, the judicious amalgamation of these hydrogels (including fibrin) with diverse materials and the application of innovative engineering techniques have emerged as promising avenues to overcome the individual limitations of these hydrogel types. This strategic synergy not only extends their potential utility in crafting intricate and functional tissues but also enriches the field of 3D printing with enhanced capabilities for tissue engineering endeavors.

### 2.3. Enhancing Tissue Engineering via ECM Bioinks: Replicating Native Microenvironments for Targeted 3D Bioprinting and Regeneration

ECM bioinks represent an innovative approach to 3D bioprinting that aims to replicate the intricate microenvironments of living tissues. These bioinks are formulated to mimic the composition and structure of the natural ECM, which is a complex meshwork of proteins, sugars, and signaling molecules that provides the foundation for cell adhesion, migration, and tissue formation. One of the major strengths of ECM bioinks is their biomimetic nature, which resembles the native tissue environment. This biomimicry offers a favorable landscape for cell interactions and influences various cellular processes, such as proliferation, differentiation, and tissue-specific functions. By incorporating familiar proteins and cues, ECM bioinks create a supportive platform for cells to thrive in vitro, ultimately promoting successful tissue integration post-implantation. ECM bioinks are promising candidates for tissue-specific regeneration. They can be sourced from tissues similar to the target organ or tissue, allowing the recreation of tissue-specific biochemical and biomechanical cues [91,92,93,94,95]. This level of customization is crucial for promoting the desired cellular responses and tissue formation, as different tissues have distinct ECM compositions that guide their development and function. However, several challenges need to be overcome. The isolation and replication of the ECM complexity are intricate processes that might involve multiple steps, potentially resulting in variations between batches. Achieving consistent and standardized ECM bioinks is an ongoing endeavor in this field. In addition, controlling the degradation rate of ECM-based constructs can be complex because natural ECM components may degrade differently than synthetic biomaterials, potentially affecting the stability and lifespan of the printed tissue [96,97,98,99].

### 2.4. Biomedical Applications through Ceramic and Metal 3D Printing: Exploring Hydroxyapatite and Titanium Alloys for Enhanced Tissue Engineering

Ceramics have emerged as a compelling option owing to their unique combination of properties. One notable ceramic is hydroxyapatite (HA), which is found in natural bones. The 3D printing process for ceramics involves the creation of a ceramic powder mixed with a binder material, allowing precise layer-by-layer deposition. The subsequent steps involve removing the binder and sintering the structure at high temperatures to achieve the desired material properties. Ceramics offer strengths that align with biomedical requirements. For instance, their high biocompatibility reduces the risk of adverse reactions to implantation. Furthermore, ceramics, such as HA resemble the mineral composition of bone, enhancing integration and tissue regeneration. Their impressive mechanical strength and stiffness render them suitable for load-bearing applications, particularly in bone tissue engineering [100,101,102,103]. However, the brittleness of ceramics can pose challenges in dynamic mechanical environments because they are prone to fracture under certain conditions. Complex manufacturing processes involving binder removal and sintering require careful parameter control [104,105]. In addition, metal alloys have garnered attention for the 3D printing of artificial blood vessels, with titanium and its alloys being prominent choices. These materials exhibit outstanding biocompatibility, corrosion resistance, and mechanical properties. This process involves the use of powdered metal particles fused layer-by-layer using heat or laser energy. One of the defining strengths of metal alloys is their ability to seamlessly integrate with the surrounding tissues, making them ideal for medical implants. Their robust mechanical strength and toughness align well with those of load-bearing applications and offer stability and durability. Moreover, the precision achieved in 3D printing using metal alloys enables the creation of intricate structures tailored to patient requirements [106,107,108]. However, the use of metal alloys for 3D printing requires certain considerations. In addition to the overall cost of the procedure, the required equipment and materials can be expensive. Post-processing steps (such as heat treatment and surface finishing) are necessary to achieve the desired mechanical properties and surface quality, thereby increasing complexity [109,110,111,112]. Additionally, it is essential to broaden the scope by considering different factors that control vascularization, such as porosity, pore size, and pore connectivity of metal/ceramic implants and coating [113]. For instance, Feng Bai et al. conducted a study exploring the impact of various pore sizes and interconnections in the production and implantation of beta-tricalcium phosphates on blood vessel formation [114]. Their findings revealed that the size of interconnections significantly influences angiogenesis. Additionally, Victor L. Correa et al. focused on investigating porosity in the process of implanting ECM-based hydrogels into metallic materials [115]. The study demonstrated network formation among endothelial cells within the hydrogel matrix and an increase in VEGF. Numerous ongoing studies aim to address these limitations, underscoring the necessity for research on their interactions to ensure both biocompatibility and functionality.

### 2.5. Advancing Regenerative Medicine: 3D Printing Living-Cell-Integrated Artificial Blood Vessels for Personalized Vascular Solutions

The innovative approach of 3D printing of artificial blood vessels through integrating living cells has gained prominence in regenerative medicine. This method involves merging bioinks, materials providing structural support, and diverse living cell types to fabricate intricate vascular structures that mirror the physiological attributes of natural blood vessels. The objective is to seamlessly align the functional blood vessels with the body’s circulatory system [116,117,118,119,120]. The choice of cells influences the success of this technique, encompassing various distinct cell types that contribute to forming functional vessels, including endothelial cells that line blood vessel interiors and play a crucial role in vascular health regulation [46,121,122,123], as well as smooth muscle cells responsible for contracting and relaxing the blood vessel walls. Integrating these cell types into printed constructs seeks to replicate the dynamic behavior exhibited by natural blood vessels [124,125,126,127]. Moreover, mesenchymal stem cells add versatility by differentiating various cell types within blood vessels, and their guided development enhances functional mimicry [128,129,130,131]. Using cells directly sourced from patients, such as induced pluripotent stem cells, enhances compatibility and reduces immune rejection risks, thus providing the potential for crafting personalized blood vessels tailored to individual patients [132,133] (Table 1).

The strengths of 3D printing of artificial blood vessels using cells are multifold. Incorporating living cells, particularly patient-derived cells, amplifies compatibility with body tissues, mitigates adverse reactions, and fosters seamless integration. Amalgamating di-verse cell types introduces complexities that mirror the behavior and functionality of native blood vessels, potentially yielding improved functional outcomes. A personalized approach that leverages patient-specific cells holds promise for crafting tailored blood vessels, reducing graft rejection risk, and enhancing long-term therapeutic results [47,128,129,130,131,132]. However, challenges remain. Maintaining cell viability throughout the 3D printing process and post-implantation is a pivotal concern that influences the success of engineered blood vessels. Establishing a functional network of blood vessels within a construct presents a complex hurdle for oxygen and nutrient exchange and overall construct viability. The intricate task of designing artificial blood vessels to closely emulate the properties and functions inherent to natural vessels represents a formidable challenge. Central to their development are the primary objectives of ensuring long-term patency, characterized by the vessel’s sustained openness, and mitigating complications, notably thrombosis, which entails the prevention of blood clot formation. Selecting a suitable bio-ink that supports cell growth, maintains structural integrity during printing, and guarantees post-implantation survival is critical [48,133,134,135]. The convergence of 3D printing and cellular biology holds a transformative potential for artificial blood vessels. It can re-define the treatments for vascular diseases, injuries, and transplantation. As technology progresses, the ability to engineer personalized functional blood vessels can impact patient care, elevate treatment strategies, and improve the overall quality of life.

### 2.6. Integrating Drugs into 3D-Printed Artificial Blood Vessels for Targeted Therapeutic Regeneration

3D printing of artificial blood vessels using drugs is an innovative approach that combines regenerative medicine and pharmaceutical technology. This method involves incorporating therapeutic agents or pharmaceutical compounds directly into 3D-printed vascular constructs, offering a unique synergy between structural support and localized drug delivery capabilities. The process begins by selecting a suitable bioink, which acts as a scaffold for the 3D printing process, and loading it with pharmaceutical agents, such as growth factors, anti-inflammatory drugs, or antibiotics [134,135]. Using a 3D bioprinter, the cell-laden bioink is deposited layer-by-layer to construct the desired blood vessel shape while simultaneously embedding the drug-loaded bioink (Figure 2). This integration of drugs serves various purposes, from promoting tissue healing through growth factors to managing inflammation with anti-inflammatory drugs and preventing infection with antibiotics. The strengths of this approach are its precise drug delivery, enhanced healing, and tailored treatment strategies. In a significant advancement, Muhammad Shafiq et al. have empirically demonstrated the augmentation of endothelialization post-transplantation through a synergistic approach. Their method involves the co-application of neuropeptide substance P (SP) alongside small-diameter artificial blood vessels constructed from PCL [136]. In another investigation, Megan Kimicata et al. harnessed gelatin methacryloyl (gelMA) laden with heparin during the fabrication of artificial blood vessels. This strategic incorporation facilitated the continuous release of heparin, leading to a robust enhancement in endothelial functionality and the establishment of an anti-thrombotic microenvironment [137]. These compelling results substantiate the tremendous potential of the engineered artificial blood vessels, rendering them more suitable for transplantation and concurrently ameliorating associated side effects. Challenges include maintaining drug stability during printing and implantation, and achieving controlled, sustained drug release from the printed constructs. This approach could transform the treatment landscape for vascular diseases, postsurgical recovery, and tissue regeneration using a new era of personalized therapeutic interventions at the intersection of regenerative medicine and pharmaceuticals [138,139,140,141,142,143].

**Figure 2 jfb-14-00497-f002:**
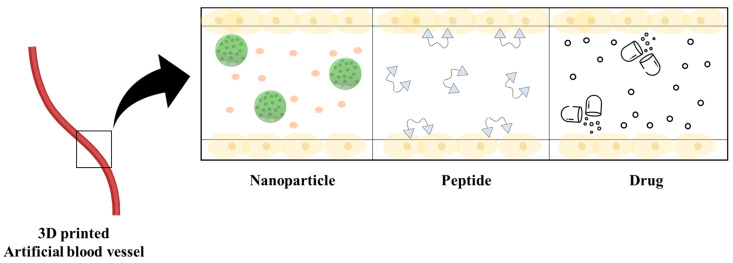
Artificial blood vessels loaded with nanoparticles, peptides, and drugs.

**Table 1 jfb-14-00497-t001:** Bioink combinations, encapsulated cell types, and applications in tissue engineering.

Bioink Combination	Encapsulated Cell Type	Applications	Ref.
Alginate, GelMA	HUVECs, Cardiomyocyte	Regenerative medicine and drug discovery applications.	[144]
Alginate, F127, gelatin	Endothelial cell	Angiogenesis and osteogenesis	[145]
Alginate, nHA, DNA, PCL	MSCs	Osteogenesis	[146]
Alginate, methylcellulose	Islets	Transplantation of pancreatic islets (Protection of transplanted islets from the immune system in diabetes type 1)	[147]
alginate/gelatin/agarose hydrogels’	iPSC	angiogenesis and nerve repair applications	[148]
Collagen, fibrin	AFSCs and MSCs	Skin wounds regeneration	[149]
Collagen, chitosan, GO-np	Chondrocytes	Cartilage protection	[150]
VdECM, alginate	EPCs	Regeneration in hindlimb ischemic disease	[134]
Gelatin, silk	Chondrocytes	Growth and proliferation of chondrocytes	[151]
GelMA, alginate, CS-AEMA	BM-MSCs	Cartilage formation	[152]
HA, PUnp	MSCs	Customized cartilage tissue engineering	[153]
OHA, hlyco chitosan, adipic acid dihydrazide	ATDC5	Cartilage regeneration	[154]

## 3. 3D Printing Methods for Artificial Blood Vessels

The development of 3D printing methods for artificial blood vessels has been an important and evolving field in tissue engineering and regenerative medicine. In the early 2000s, extrusion-based bioprinting, characterized by controlled deposition of bioinks through a nozzle, emerged as one of the initial 3D printing methods for tissue engineering [155,156]. Researchers primarily employed this method to fabricate basic tubular structures as precursors to artificial blood vessels [157,158]. However, the challenges during this period centered around optimizing bioink compositions for vascular tissue and ensuring precise cell placement [157,159]. In the mid-2000s, inkjet bioprinting, inspired by traditional inkjet printing technology, gained prominence for creating vascular constructs [160,161]. It facilitated precise and rapid droplet deposition of cell-laden bioinks. Researchers devised techniques to safeguard cell viability and adjust droplet sizes, enhancing control over tissue microstructures [162]. In the late 2000s, Stereolithography (SLA) bioprinting, which utilizes photopolymerization to construct 3D structures layer by layer, was adapted for tissue engineering purposes [163,164,165]. Researchers delved into its potential to fashion intricate vascular networks by fine-tuning the printing process. However, the challenges during this era revolved around the selection of biocompatible resins and the optimization of printing parameters for blood vessel-like structures [166,167]. In the early 2010s, Selective laser sintering (SLS) bioprinting, distinguished by its material diversity, garnered attention in tissue engineering, including the creation of blood vessel constructs [166,168,169,170]. Researchers explored the use of biocompatible powders and laser sintering to craft complex structures. However, the challenges included ensuring the biocompatibility of SLS materials and reproducing the intricacies of the vascular microenvironment [170,171]. In the 2010s, electrospinning bioprinting, specializing in the production of nanofibrous scaffolds, garnered interest in supporting vascular tissue engineering [172,173] (Figure 3). Researchers devised methods for generating nanofibrous scaffolds that emulate the ECM and encourage the growth of endothelial cells [174,175]. Integration with complementary techniques aimed to augment the complexity of vascular networks. Throughout this timeline, advancements in materials, bioinks, and printing technologies bolstered the biocompatibility and precision of artificial blood vessels. The advancements in bioprinting technology have provided solutions to numerous challenges and have successfully achieved the goal of emulating the structural and physiological characteristics of native blood vessels. The development of bioprinting has enabled the emulation of diverse and intricate vascular networks within the body, including microvasculature, and has resulted in vascularization through artificial blood vessels upon transplantation. Furthermore, by supporting the growth of endothelial cells and smooth muscle cells in the vicinity of the transplant site and regulating inflammatory responses, these bioprinted structures can perform endothelial functions. Thus, artificial blood vessels that mimic features akin to those found in the body can fulfill roles such as nutrient supply, blood flow regulation, prevention of blood clotting, long-term survival of endothelial cells, and preservation of vascular functionality within engineered structures [48,176,177,178]. While each method possessed unique strengths and faced its own set of challenges, researchers continued to refine and amalgamate these techniques to craft functional and biocompatible blood vessel constructs for applications in tissue engineering and regenerative medicine [179] (Figure 4).

### 3.1. Extrusion-Based Bioprinting in 3D Printing Methods for Artificial Blood Vessels

Extrusion-based bioprinting is a key 3D printing technique for creating artificial blood vessels. This method involves depositing bioink layer-by-layer through a nozzle and adapting traditional 3D printing for biological materials, such as cells and biomaterials [120,180,181,182,183]. The bioink was designed to mimic the ECM of natural tissues and provide a scaffold for cellular growth, organization, and vascular structure formation. The process begins with formulating the bioink and determining its viscosity, mechanical properties, and cell compatibility. This typically includes hydrogels, polymers, and other biomaterials that provide mechanical support while enabling cell encapsulation [184,185,186]. If cells are used, they are mixed with the bioink. The bioprinter nozzle deposits the bioink layer-by-layer onto a substrate following precise instructions from a computer-controlled process. After deposition, cross-linking agents or ultraviolet (UV) light are used to solidify the bioink and maintain its integrity. Extrusion-based bioprinting offers advantages, such as preserving cell viability during printing, supporting various cell types and biomaterials, and enabling large-scale constructs with reasonable resolution [183,187,188]. However, challenges include balancing the print speed with cell viability, selecting suitable bioink properties, and developing functional vascular networks within the construct. Extrusion-based bioprinting involves loading bioink into a piston and distributing it using either pneumatic pressure or mechanical (piston or screw-driven) methods. Piston-driven systems provide more direct control over the flow of bioink, whereas pneumatic-based systems offer greater spatial control and can distribute bioink with higher viscosity [120,183,189]. However, this extrusion method can generate higher pressure forces, potentially leading to cell damage for loaded cells. Moreover, high printing speeds cause mechanical stress on the cells; if printing speeds are too high, the shear forces created during extrusion can damage delicate cells or disrupt tissue within the printed structure. This can result in reduced cell viability and impaired tissue function. Additionally, when distributing high-viscosity bioinks, issues such as reduced cell viability due to shear stress and nozzle clogging can [189,190,191]. To address these challenges, researchers have focused on modifying parameters such as the printing speed, dispensing pressure, nozzle geometry, and properties of bioinks to overcome nozzle clogging and enhance cell viability [192]. Slower print speeds can reduce shear stress and allow for better cell survival. However, finding the right balance between speed and printing efficiency remains a key challenge. Furthermore, the development of gel-in-gel bioprinting techniques has shown promise. In this approach, bioink serves as a support material, being extruded within a hydrogel matrix, enabling self-healing properties [193]. Creating functional vascular networks within bioprinted constructs is crucial for delivering nutrients and oxygen to cells throughout the tissue. Without proper vascularization, thicker constructs may experience limited cell viability in their inner regions due to inadequate nutrient and oxygen diffusion. Researchers are exploring strategies such as incorporating endothelial cells into bioinks to promote blood vessel formation (angiogenesis) or using sacrificial materials that can be removed post-printing to create hollow channels for vasculature [134].

### 3.2. Inkjet Bioprinting in 3D Printing Methods for Artificial Blood Vessels

Inkjet bioprinting is a refined and sophisticated technique in the field of 3D printing that is specifically tailored to create intricate artificial blood vessels. Drawing inspiration from conventional inkjet printing, this method adapts the technology to handle biological materials. This allows for the precise layer-by-layer deposition of bioink droplets, ultimately constructing complex and detailed vascular architectures. The workflow begins with the formulation of a bioink designed to serve as a carrier for living cells, biomaterials, and therapeutic agents. This bioink is optimized for properties, such as viscosity and gelation behavior to facilitate accurate deposition while maintaining cellular viability and growth support [194,195,196]. Using computer-aided design (CAD) software 24.0, a digital model of the desired blood vessel structure is created, detailing the dimensions, branching patterns, and architectural features. The printhead of the bioprinter, equipped with microscopic nozzles, ejects minute droplets of the bioink onto a substrate in a controlled manner. These droplets accumulate layer-by-layer, gradually forming a 3D structure of the artificial blood vessel. The precision of this process ensures that each droplet adheres to the preceding layer, thereby producing a coherent and integrated final construct. Solidification of the bioink is achieved through methods, such as UV light exposure or crosslinking agents, preserving the stability and form of the bioink. The advantages of inkjet bioprinting include its ability to achieve high-resolution printing, gentle handling of living cells, and the potential for automated production [197,198,199,200]. High resolution is crucial for creating intricate tissue architectures and mimicking the complexity of natural tissues. Inkjet bioprinters can produce droplets of bioink, and the size of these droplets is a key factor in achieving the desired resolution. Droplets in the picoliter (10^−12^ L) to nanoliter (10^−9^ L) range are common in inkjet bioprinting. Droplets of the order of a few picoliters can result in precise and fine printing, allowing for the creation of detailed tissue structures [201]. In tissue engineering, it is often desired to achieve a resolution of the order of 100 μm or smaller to accurately replicate the fine details of tissues and cellular structures. The desired resolution for inkjet bioprinting can vary depending on the specific application but typically falls in the range of micrometers to hundreds of micrometers [199,202]. The challenges include the formulation of an optimal bioink, limited biomaterial options compatible with the printhead, and ensuring strong layer-to-layer adhesion.

### 3.3. SLA in 3D Printing Methods for Artificial Blood Vessels

SLA stands as a prominent technique used in 3D printing for creating intricate artificial blood vessels. This method capitalizes on the unique characteristics of photosensitive liquid resins to build complex vascular structures layer-by-layer, offering a controlled and precise approach to tissue engineering [203,204,205,206]. The workflow begins with the careful selection of a photosensitive liquid resin that can solidify under UV light exposure, forming the basis of the 3D structure. A digital model of the desired blood vessel design was generated using CAD software. This digital blueprint outlines dimensions, branching patterns, and architectural intricacies. During the printing process, a platform is submerged in liquid resin, and a laser or UV light is employed to cure the resin layer by layer. As the resin solidifies, it transforms from a liquid to a solid state, forming a scaffold for an artificial blood vessel. This technique offers distinct advantages in terms of its high precision, smooth surface finish, and potential to incorporate various biomaterials. High precision is vital for accurately replicating the intricate and organized nature of blood vessels, which have specific diameter variations and branching patterns. Blood vessels in the body have smooth inner surfaces to ensure the efficient flow of blood. The ability of SLA bioprinting to produce smooth surfaces can help mimic the natural environment of blood vessels, reducing the risk of clot formation and promoting healthy blood flow. SLA bioprinters can work with various biomaterials, including those that are biocompatible and suitable for creating blood vessel constructs. This versatility enables the incorporation of materials that promote tissue regeneration and vascularization. SLA bioprinting can be used to create a wide range of structures, from small tissue constructs to larger organ models. The desirable size of the printed structures can vary significantly depending on the specific application. In regenerative medicine and tissue engineering, small tissue constructs with dimensions of the order of centimeters or millimeters may be suitable for in vitro testing. SLA bioprinting is known for its high resolution, allowing for the precise fabrication of intricate structures. The desirable resolution typically falls within the micrometer (μm) range. Achieving sub-100 μm resolution is often important for replicating the fine details of tissue microstructures, such as cellular arrangements and vascular networks. SLA bioprinting can be used to create microfluidic channels and structures with sub-10 μm features for applications in drug testing and diagnostics. When speed is a priority and high resolution is not critical, a higher layer thickness (200–500 μm) may be used to expedite printing [163,206,207,208]. Challenges include ensuring the biocompatibility of the chosen resin and addressing the curing depth for uniform and complete solidification. SLA is a sophisticated avenue for 3D printing to construct detailed artificial blood vessels [203,205,209,210,211,212]. SLA offers precision and surface quality by using a photosensitive liquid resin and precise UV light or laser exposure. The ability of this method to fabricate intricate structures holds potential for advancing regenerative medicine and vascular therapies, although careful consideration of biocompatibility and curing depth remains crucial.

### 3.4. SLS in 3D Printing Methods for Artificial Blood Vessels

SLS is a pivotal technique within the spectrum of 3D printing methods used to create intricate artificial blood vessels. This method uses high-energy lasers and powdered biomaterials to create complex vascular structures, offering versatility in terms of material composition and structural intricacies [116,213,214]. The workflow begins with the selection of powdered biomaterials that serve as the foundation for the construction of artificial blood vessels. The platform is coated with a layer of powdered biomaterial, and a high-energy laser is employed to selectively fuse or sinter the particles, forming a solid layer adhering to the platform and previous layers. This process is repeated layer-by-layer to progressively build a 3D structure. The strength of SLS lies in its material diversity, which enables the use of a wide range of biomaterials, and its capacity to fabricate intricate geometries. However, challenges include the precise control of laser parameters for accurate sintering, handling of powdered biomaterials for uniform distribution, and achieving the desired surface finish [215,216,217]. SLS may have shortcomings compared to other techniques, but with innovative approaches and applications, we believe it holds the potential to advance the capabilities of artificial blood vessels. While SLS may not be suitable for directly fabricating blood vessels for implantation, it can still play a role in supporting vascular tissue engineering research and development. For example, SLS can be used to create molds, scaffolds, or support structures for biofabrication processes that utilize biocompatible materials and techniques better suited for blood vessel production, such as bioprinting or electrospinning. There have been reports of research overcoming these limitations by using SLS to fabricate scaffolds that promote bone regeneration and vascularization. These instances can be considered as cases where SLS’s drawbacks have been surmounted [116,214]. Moreover, research has shown successful transplantation of artificial blood vessels produced using the SLS technique in animal models. In one study, a C-shaped ring was fabricated using a biodegradable polyester material via SLS. When this ring was deposited onto the transplant’s poly(glycerol sebacate) prepolymer, it not only enhanced the tensile strength of the artificial blood vessel but also provided structural support. Mechanical testing demonstrated that the PCL ring endowed the transplant with the appropriate elasticity, preventing collapse and excessive deformation of the graft wall. Additionally, the transplanted artificial blood vessel exhibited excellent vascularization [213]. Despite some challenges, SLS presents a promising avenue for advancements in regenerative medicine, combining the fabrication of artificial blood vessels and more complex structures with a diversity of biomaterials.

### 3.5. Electrospinning in 3D Printing Methods for Artificial Blood Vessels

Electrospinning is a unique 3D printing approach for creating artificial blood vessels. Although electrospinning is not a conventional 3D printing technique, it plays a crucial role in the fabrication of nanofibrous scaffolds that contribute to the construction of blood vessel structures in tissue engineering. This process involves the preparation of a polymer solution, which is then subjected to a high-voltage charge. This charge causes the polymer solution to form a fine jet that stretches and solidifies into nanofibers as it is collected on the substrate. This results in a nanofibrous scaffold that mimics the intricate architecture of natural tissues. Additionally, Electrospinning is commonly used to create nanofibrous scaffolds that can serve as a framework or template for blood vessel regeneration. These scaffolds can mimic the ECM of blood vessels and provide structural support for cell attachment, proliferation, and tissue formation. Moreover, Electrospun nanofibers can promote the adhesion and growth of endothelial cells, which form the inner lining of blood vessels. This is crucial for the development of functional blood vessels [172,173,218]. Additionally, Electrospun scaffolds can incorporate bioactive molecules or drugs that aid in angiogenesis (the formation of new blood vessels) and support tissue healing [219]. However, although electrospinning can create fine-scale structures, replicating complex vascular networks found in the body, with interconnected vessels, is challenging. This is due to the limitations in creating controlled, branched structures at the microscale and the requirement for continuous lumens (inner channels) for blood flow. To create functional blood vessels, Electrospinning is often combined with other techniques such as bioprinting or microfabrication. These techniques can complement each other to achieve the desired complexity and functionality [220,221]. Regarding size limitations, Electrospinning can produce nanofibers with diameters typically in the range of tens to hundreds of nanometers. These fine fibers can mimic the scale of natural ECM components. The dimensions of Electrospun scaffolds can be controlled based on the design of the Electrospinning setup and the collection method. Scaffolds can be fabricated in various shapes and sizes, including sheets, tubes, or three-dimensional constructs [222,223]. The strengths of electrospinning include its ability to create nanofibrous structures, biocompatibility with living cells, and fine control over scaffold properties. The challenges include achieving 3D complexity, maintaining scaffold integrity, and scaling up the process for larger constructs [170,171,224,225]. Despite these challenges, electrospinning remains a valuable tool in tissue engineering, contributing to the advancement of artificial blood vessel creation and other applications in regenerative medicine.

### 3.6. Post-Production Process

Once each layer of the artificial blood vessel was deposited, the step of crosslinking or solidification was initiated to ensure the shape and integrity of the printed structure. This involves activating mechanisms that bond biomaterials and provide stability to the construct. Upon completing the printing phase, the construct transitions to a series of post-production processes aimed at refining its characteristics. Washing removed any residual bioink or chemicals, ensuring the purity and suitability of the construct for the subsequent stages. Maturation involves the creation of an environment conducive to cellular interactions and organization within the construct. This step is pivotal for developing functional and physiologically relevant artificial blood vessels. Then, the construct is rigorously tested and validated. Various assessments have been conducted to evaluate the mechanical properties, structural robustness, and overall functionality of the printed blood vessels [180,226,227,228,229,230]. Additionally, regarding materials, an examination is conducted to determine whether biodegradable polymers and similar materials have been used for the fabrication of biocompatible vascular scaffolds. These materials are well-tolerated by the body and gradually degrade over time as natural tissue forms [231,232]. Components of the natural ECM, such as collagen and gelatin, are employed in creating vascular scaffolds, as they can promote cell adhesion and tissue integration, which are also evaluated [233,234]. Hydrogels such as alginate, fibrin, and hyaluronic acid provide a hydrated environment resembling that of natural tissue. They encapsulate cells and promote cell survival and growth, making them subjects of evaluation, particularly in assessing cell viability and survival [235,236,237]. Furthermore, by nature, the materials used for bioprinting blood vessels must exhibit high biocompatibility to minimize adverse reactions when implanted into the body. They should not induce inflammation, thrombosis, or immune responses [48]. Printed blood vessels should possess mechanical properties similar to those of native blood vessels, including elasticity and strength. These properties are crucial for withstanding blood pressure and maintaining vascular function [119,180]. Successfully bioprinted blood vessels often exhibit the ability to support the growth and alignment of endothelial cells, which form the inner lining of blood vessels. This endothelialization is essential for proper blood flow and preventing clot formation. Bioprinted blood vessels should support angiogenesis. This ensures adequate blood supply to the surrounding tissues. The bioprinted blood vessels should remain open and patent, without blockages or narrowing, to ensure uninterrupted blood flow [50,238,239].

## 4. Applications of 3D Bioprinted Artificial Blood Vessel

### 4.1. Harnessing the Potential of 3D-Printed Artificial Blood Vessels in Medical Advancements

The convergence of innovative technology and medical innovation has led to a remarkable breakthrough in health care: artificial blood vessels have been crafted using advanced 3D printing techniques. These constructs hold promise across a spectrum of medical applications, revolutionizing approaches to surgical interventions, organ transplantation, tissue engineering, and regenerative medicine [47,240,241,242]. One of the most significant contributions of 3D-printed artificial blood vessels is surgical revascularization. Patients with conditions, such as atherosclerosis, often face the daunting challenge of narrowed or blocked arteries. In such cases, skilled surgeons can harness engineered blood vessels to bypass or replace the damaged vasculature. This approach restores proper blood flow, dramatically mitigating the risk of catastrophic events, such as heart attacks and strokes. By integrating these advanced constructs into the intricate network of the circulatory system, surgeons can empower themselves with innovative tools to combat vascular ailments. Artificial blood vessels are indispensable in the intricate landscape of organ transplantation. The viability and success of transplanted organs depend on a robust blood supply to sustain the transplanted tissue. Here, 3D-printed blood vessels emerged as pivotal facilitators, seamlessly melding with the transplanted organ to ensure optimal perfusion. This integration enhances the chances of transplant success, while simultaneously diminishing organ rejection. The prospects of successful organ transplantation are augmented by integrating these engineered constructs, providing newfound hope for patients in dire need of life-saving interventions [48,243,244,245,246]. Tissue engineering, an expanding frontier in medical science, welcomes game-changing components of functional artificial blood vessels. The development of complex tissues and organs in laboratory settings benefits from the integration of these constructs. Engineered to replicate the vasculature of natural tissues, these blood vessels serve as lifelines, supplying crucial nutrients and oxygen to growing tissues. As scientists endeavor to engineer intricate tissues for both research and therapeutic applications, these constructs have emerged as enablers, nurturing the development and maturation of intricate tissues that hold promise for an array of medical advancements. Shen et al. pioneered the development of vascularized tissue for bone regeneration using a combination of gelatin and PLA-PEG through an extrusion-based 3D bioprinting technique. Their groundbreaking research, published in 2018, demonstrated significantly improved recovery outcomes in the vascularized scaffold when both bone marrow stromal cells (BMSCs) and vascular endothelial cells were co-printed. This advancement outperformed the group in which only the scaffold was implanted, under-scoring the critical role of vascularization in enhancing tissue regeneration [244]. Furthermore, in 2018, Arai et al. engineered a cardiac structure using a combination of fibroblast cells and induced pluripotent stem cell-derived endothelial cells. However, it is essential to note that this approach has limitations in generating whole organs or large-sized tissues due to the absence of vascular structures during the printing process [247]. In a parallel effort in the same year, Maiullari et al. achieved a significant milestone by fabricating cardiac tissue in vitro. Their method involved the use of xenografts composed of induced pluripotent cell-derived cardiomyocytes (iPSC-CMs) and human umbilical vein endothelial cells (HUVECs). This innovative approach encapsulated cells within a hydrogel containing alginate and PEG-fibrinogen (PF) and utilized a microfluidic printing head (MPH) for high-resolution spatial printing. Notably, the bioprinted cardiac tissue products featured intricate vascularized networks. Subsequent in vivo implantation studies revealed that these vascularized networks significantly promoted the integration of the host’s vasculature with the fabricated products, highlighting their potential for enhancing tissue integration and function [248].

In summary, the union of advanced 3D printing technology and medical science has led to a paradigm shift in artificial blood vessels. These constructs represent a brighter and healthier future. As medical frontiers continue to expand, the role of 3D-printed artificial blood vessels is a testament to human ingenuity, ushering in an era in which medical challenges are met with innovative solutions that bridge the gap between possibility and reality.

### 4.2. Utilizing 3D-Printed Artificial Blood Vessels to Study Drug Interactions, Disease Modeling, and Precision Medicine

In medical research and drug development, artificial blood vessels crafted using advanced 3D printing techniques are emerging as potent tools for revolutionizing the landscape of drug testing and research. These engineered constructs have the potential to elucidate the intricate interactions between drugs and blood vessels, paving the way for novel therapies and treatments targeting a spectrum of vascular diseases and disorders.

Artificial blood vessels offer researchers a designed and controlled environment in which to investigate the effects of drugs and treatments on the vascular system. This controlled platform is paramount for deciphering the complex interplay between pharmaceutical agents and blood vessels and shedding light on potential outcomes and responses. Researchers can employ artificial blood vessels to model various disease pathways and conditions, mimicking the physiological and pathological processes within the vasculature. By inducing specific disease conditions within these constructs, scientists can observe how drugs interact with the diseased blood vessels, thereby identifying potential therapeutic interventions and elucidating their underlying mechanisms [235,249,250,251,252,253].

Therefore, the efficacy and safety of potential pharmaceutical candidates are of paramount importance. Artificial blood vessels provide testing grounds for evaluating the effects of drugs on blood vessel function, permeability, and overall health. This insight aids in validating drug efficacy and safety profiles and offers a reliable preclinical platform before transitioning to human trials. The advent of precision medicine calls for tailored treatments that consider individual variations in the response to drugs. Artificial blood vessels, combined with patient-derived cells, are open avenues for personalized medical approaches. Integrating artificial blood vessels expedites drug discovery by providing a versatile and physiologically relevant platform for initial testing. This has accelerated the identification of potential drug candidates with promising effects on vascular health, streamlining the path toward clinical trials and patient care. By harnessing these engineered constructs, researchers can investigate the intricate molecular and cellular mechanisms underlying drug interactions with blood vessels. This deeper understanding is invaluable for uncovering novel targets for therapeutic interventions and allowing the development of drugs that precisely target specific pathways and components within the vasculature. In a groundbreaking study, Njoroge et al. achieved a significant milestone in tissue engineering by producing human blood vessels using the Electrospinning technique. These tissue-engineered blood vessels (TEBV) were subjected to perfusion with fluorescently labeled human platelets and endothelial progenitor cells (EPCs). Their responses were meticulously monitored in real-time using fluorescence imaging. Interestingly, the study delved into the effects of certain pharmaceutical agents on TEBV function. It was observed that ketamine, a commonly used anesthetic in in vivo models, significantly inhibited platelet aggregation within injured TEBV. Furthermore, Atorvastatin, a cholesterol-lowering medication, was found to enhance EPC attachment to damaged TEBV. This innovative approach utilizing TEBV provides a potent and versatile alternative to existing in vivo drug testing models, allowing for the comprehensive evaluation of thrombus formation and EPC recruitment upon the infusion of human blood or blood components under physiological conditions [254]. Additionally, in a study led by Musa et al., a novel technique was introduced to quantitatively assess the anti-aggregation and anti-thrombotic properties of various three-dimensional vascular structures fabricated using additive manufacturing methods. This innovative methodology leveraged real-time measurements of cytosolic Ca^2^⁺ signals to evaluate platelet activation in suspensions of fluorescently labeled human platelets through fluorescence spectrofluorescence. Simultaneously, fluorescence imaging of DiOC6-labeled platelets was employed to visualize thrombus formation on the surface of these engineered constructs [255]. The experiments conducted using this method yielded insightful results. For instance, it was discovered that type I collagen hydrogels, frequently employed as scaffolds in vascular tissue engineering, exhibited limited capacity to induce significant platelet activation. In contrast, type I and type III neocollagen secreted from human coronary artery smooth muscle cells displayed superior compatibility with these hydrogels and supported thrombus formation within the medial layer. Intriguingly, the inclusion of an intimal layer composed of human umbilical vein endothelial cells over the medial layer proved effective in inhibiting platelet activation and aggregation. This methodology showcased the ability to quantitatively compare the thrombotic potential of different vascular structures, making it a valuable tool for the standardized assessment of functional properties in tissue-engineered vascular constructs developed using various culture techniques [255].

In the ever-evolving medical science landscape, artificial blood vessels are dynamic tools that transcend traditional research methods. By offering a controlled environment to explore drug interactions, model disease conditions, validate drug efficacy and safety, and unravel underlying mechanisms, these constructs are useful in a new era of precision in drug testing and research. As we continue to unravel the mystery of vascular health and disease, the synergy between advanced 3D printing techniques and pharmaceutical investigations holds promise for transformative breakthroughs in patient care and therapeutic advancement.

### 4.3. Exploring Uncharted Territory: Probing New Blood Vessel Research and Disease Onset with Artificial Constructs

The emergence of artificial blood vessels using innovative 3D printing technologies has opened an exciting avenue for groundbreaking research in the realms of blood vessel bi-ology and disease initiation. These innovative constructs not only mimic the intricacies of natural blood vessels but also provide a versatile platform to unravel the mystery of vascular development, disease initiation, and potential therapeutic interventions.

Artificial blood vessels present an opportunity to venture into unexplored territories of blood vessel research. By designing and fabricating these constructs, scientists can investigate the mechanics of blood vessel behavior by studying factors, such as blood flow dynamics, endothelial cell interactions, and vessel elasticity. This exploration provides novel insights into the function and regulation of blood vessels, revealing pathways to optimize cardiovascular health. Engineered blood vessels serve as remarkable tools for simulating the onset and progression of vascular diseases. Researchers can recreate disease-inducing conditions, such as plaque deposition in atherosclerosis, dysregulation of blood pressure in hypertension, or weakening of vessel walls in aneurysms. By observing the manifestation of these conditions in artificial vessels, scientists can decipher the mechanisms that drive disease initiation and progression.

In addition, artificial blood vessels hold promise in tissue engineering and regenerative medicine. Researchers can experiment with novel biomaterials, growth factors, and cellular components to engineer functional blood vessel substitutes. These constructs could revolutionize vascular grafts for transplantation—offering solutions for patients requiring vascular repair due to trauma, congenital anomalies, or disease. The applicability of artificial blood vessels extends beyond the limitations of vascular research. These constructs can be integrated into broader studies involving organ-on-a-chip systems, thereby enabling a holistic understanding of how blood vessels interact with various organs and tissues. This interdisciplinary approach has the potential to revolutionize our under-standing of complex physiological processes. Current models used for studying blood vessel behavior have several limitations. Many of these models are simplified representations that lack the intricate three-dimensional structure and cellular diversity found in actual blood vessels. Moreover, they often fail to account for the dynamic nature of blood flow and the mechanical forces acting on blood vessels, which are crucial factors in understanding their behavior. Additionally, existing models may not fully replicate disease-inducing conditions, such as the progressive buildup of plaque in atherosclerosis or the mechanical changes in aneurysms. Furthermore, these models may not adequately simulate the complex interactions between endothelial cells, smooth muscle cells, and other cell types within blood vessel walls. Bioprinting offers a promising avenue to address these limitations in current models. With bioprinting, researchers can create highly realistic, three-dimensional models of blood vessels that closely resemble native tissue. These bioprinted models can incorporate various cell types and extracellular matrix components, enhancing their realism. Importantly, bioprinted models can also be dynamic, incorporating fluid channels to simulate blood flow dynamics and the effects of shear stress on endothelial cells. This dynamic aspect provides a more accurate representation of blood vessel behavior. Bioprinting also enables the precise recreation of disease conditions, such as plaque deposition in atherosclerosis or vessel wall weakening in aneurysms, making these models more effective for studying disease processes. Furthermore, bioprinting allows for the controlled placement of different cell types, facilitating the study of complex cell-cell interactions within blood vessels. In recent examples of organ-on-a-chip technology, researchers have successfully utilized bioprinting to create heart-on-a-chip and vasculature-on-a-chip platforms, enabling the study of cardiac physiology, drug responses, and vascular diseases with greater precision and physiological relevance [235,249,250,251,252,253,254,255].

In summary, the advent of artificial blood vessels using advanced 3D printing methodologies has unlocked a tremendous number of research opportunities. By investigating novel blood vessel behaviors to decode disease initiation, from testing therapies to engineering functional replacements, these constructs offer a versatile canvas for researchers to paint new knowledge. As the boundaries of scientific exploration expand, artificial blood vessels are becoming powerful tools for transforming our understanding of cardiovascular health, disease mechanisms, and avenues for therapeutic advancement.

## 5. Conclusions and Future Perspectives

The convergence of innovative 3D printing technology and the intricate world of vascular biology has given birth to groundbreaking advancements in artificial blood vessels. These engineering and biological marvels have transformative potential across a spectrum of medical applications, offering novel solutions to age-old challenges and opening doors to unprecedented research opportunities. While the present achievements are commendable, the true power of artificial blood vessels lies on the horizon where they can redefine the landscape of health care.

The synthesis of biomaterials, bioinks, and living cells using sophisticated 3D printing techniques has resulted in artificial blood vessels that mimic the form and function of native vessels. In the field of artificial blood vessel production through 3D printing, the primary objective is to create transplantable microenvironments mimicking the structural complexity of native blood vessels. Natural human arteries exhibit a three-layered structure consisting of the intima, media, and adventitia, intricately intertwined to facilitate blood circulation. These structural elements are closely associated with the physiological functions of blood vessels. In particular, the intima must provide an open pathway for blood flow, with vessel sizes ranging from the microscale of capillaries (5 μm) to the macroscopic dimensions of the aorta (30 mm). To achieve the intricate microstructures and hollow lumens required for these artificial vessels, bioprinting techniques such as extrusion-based and inkjet-based bioprinting employing core-shell nozzles are considered suitable. Currently, synthetic polymers represent the minimum requirement for materials capable of withstanding the mechanical stress and pressure exerted by blood flow. It has been established that relying solely on natural materials often falls short in terms of mechanical strength. Further research endeavors should prioritize identifying optimal methods for blending synthetic polymers with natural materials, capitalizing on their complementary characteristics to advance the field of artificial blood vessel production. In 2017, Ge Gao et al. achieved a remarkable feat by fabricating three-dimensional (3D) artificial blood vessels incorporating nanoparticles. Their groundbreaking work represented a significant advancement in vascular engineering. Notably, these artificial blood vessels, created using advanced 3D printing technology, exhibited vastly superior characteristics when compared to groups transplanted solely with conventional vascular endothelial cells. This achievement exemplifies the potential of hybrid structures that seamlessly integrate cutting-edge nanotechnology, biomaterials, 3D printing techniques, and live cells, promising revolutionary advancements in the field of tissue engineering and vascular research [134]. Furthermore, Maiullari et al. achieved a major milestone in tissue engineering by fabricating cardiac tissue in vitro. Their pioneering methodology involved the utilization of xenografts composed of iPSC-CMs and HUVECs. Employing an innovative approach, they encapsulated these cells within a hydrogel matrix containing alginate and PF, precisely applying them using an MPH to create high-resolution spatial patterns. Notably, the resulting cardiac tissue products exhibited intricate vascularized networks. Subsequent in vivo implantation studies unveiled the remarkable capacity of these vascularized networks to significantly enhance integration with the host’s vasculature, under-scoring their potential to elevate tissue integration and overall functionality to new heights [248]. Similar to the pioneering developments highlighted above, the advancement of technologies aimed at printing vascularized tissue and seamlessly integrating various cutting-edge methodologies such as nanotechnology, 3D printing, and regenerative materials is poised to emerge as the cornerstone of next-generation innovations in the field. The potential of these technologies is very remarkable, promising a future where the creation of artificial blood vessels and complex tissue structures becomes not only achievable but also highly efficient and effective. This evolving landscape represents a compelling frontier in biomedical engineering, holding the promise of transformative breakthroughs that will undoubtedly shape the future of regenerative medicine and tissue engineering.

3D printing offers numerous advantages in the realm of biocompatibility. It enables precise customization of medical implants and tissue scaffolds, accommodating individual patient needs and complex anatomical structures. A wide range of biocompatible materials, including metals, ceramics, and biodegradable polymers, can be used for 3D printing. This customization and material flexibility promote a better fit for patients and reduce the risk of infection due to the seamless structures. In clinical transplantation, 3D printing offers remarkable benefits. It empowers the creation of patient-specific materials, enhancing transplantation outcomes by ensuring a seamless fit and reducing surgery time. Notably, 3D printing finds applications in orthopedics, dental implants, and tissue engineering, with successful transplants demonstrating improved patient comfort and reduced complications. Nonetheless, this innovation faces regulatory hurdles and necessitates long-term studies to assess product durability and integration with the patient’s body.

Surgical revascularization has gained newfound precision in reducing the burden of heart diseases and stroke. Organ transplantation is reliable for engineered blood vessels, enhancing graft viability, and reducing the risk of rejection. Tissue engineering has ventured into uncharted territories by incorporating functional vasculature, thereby enabling the growth of complex tissues and organs. Regenerative medicine gains traction as damaged vessels are rejuvenated, fostering healthier circulation and improving patients’ quality of life. Therefore, artificial blood vessels are more promising. Precision medicine can flourish through patient-specific constructs, mitigating immune reactions, and enhancing long-term therapeutic outcomes. Integrating therapeutic agents can turn engineered blood vessels into conduits for targeted drug delivery and revolutionize disease management through controlled release mechanisms. In addition to cardiovascular applications, artificial blood vessels may accelerate the development of functional organs, thereby propelling regenerative medicine to new heights. These constructs offer an experimental platform for investigating disease mechanisms and potentially reshaping disease management by emulating disease conditions. Collaborations across fields, such as 3D printing, nanotechnology, and biomaterial science can yield hybrid constructs with exceptional properties, while ethical considerations and regulatory frameworks must ensure patient safety and responsible innovation in this evolving landscape.

## Figures and Tables

**Figure 1 jfb-14-00497-f001:**
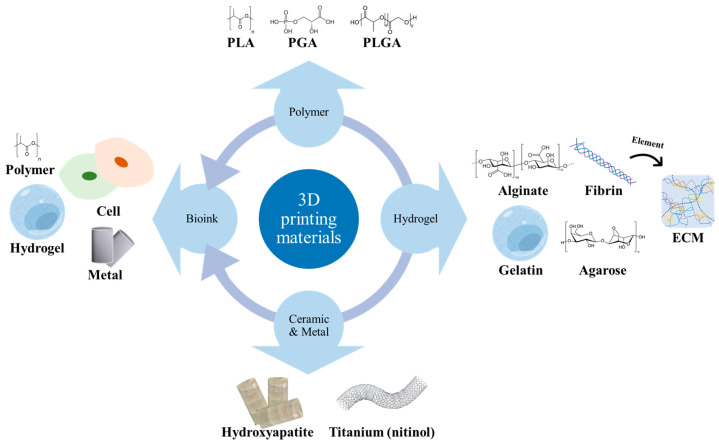
Types of representative bioinks used in 3D printing.

**Figure 3 jfb-14-00497-f003:**
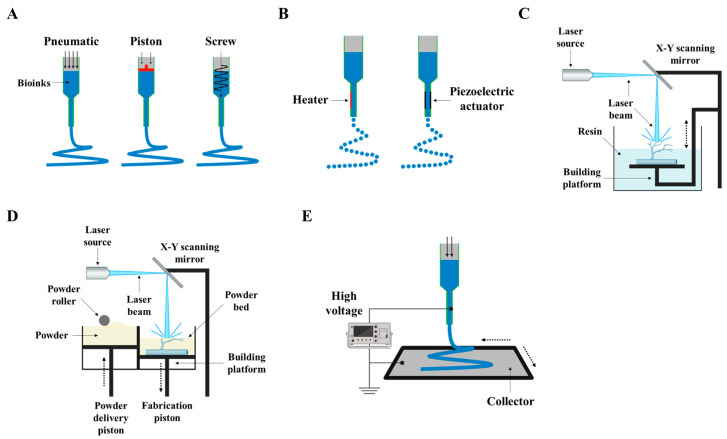
Overview of diverse 3D printing methods illustrated schematically. (**A**) Extrusion-based bioprinting, (**B**) inkjet-based bioprinting, (**C**) stereolithography (SLA), (**D**) selective laser sintering (SLS), and (**E**) electrospinning.

**Figure 4 jfb-14-00497-f004:**
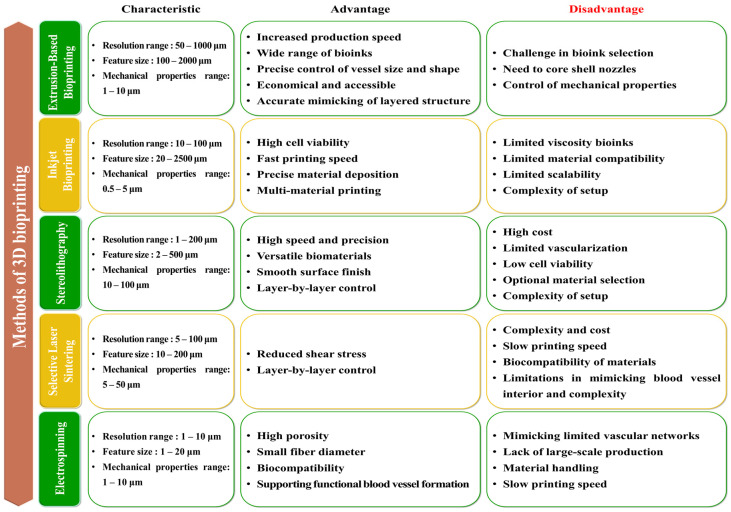
Comparative analysis of the advantages and disadvantages of various 3D printing methods.

## Data Availability

The data used to support the findings of this study are included in the article.

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
