# Peer review of "Development of Biocompatible 3D-Printed Artificial Blood Vessels through Multidimensional Approaches"

_jfb, 2023, doi:10.3390/jfb14100497_

Round 1

Reviewer 1 Report

The authors have selected an interesting subject for providing a state of the art analysis, but the manuscript in its current state does not do justice to the subject.

1. The introduction part does not properly cover the different aspects of 3D printed blood vessels, and discusses 3D printing in a very generic way.

2. What are the specific requirements in 3D printing of blood vessels? How is it different from 3D printing of other biological organs?

3. Section 2.4 of the manuscript presents discussion on ceramic and metal 3D printing. How are these related to 3D printing of blood vessels?

4. Similarly, section 3.4 deals with SLS. Can it be a 3D printing method for artificial blood vessel printing?

5. The article does not present a clear idea about which type of materials and processes are more suitable for 3d printing of blood vessels, what are the challenges, and what is the way forward.

The quality of english language is acceptable.

Author Response

We wish to submit our revised manuscript (Review Article, No. jfb-2631892) entitled “Development of Biocompatible 3D Printed Artificial Blood Vessels through Multidimensional Approaches” to be considered for publication in Journal of Functional Biomaterials. At first, the authors would like to thank the editor and reviewers for the kind and valuable comments. The authors have carefully read the editor and reviewers' comments and made a point-by-point response to the reviewers (please refer to the answer sheet). We believe that this paper is underestimated, and we have added a series of supplements, revisions, and explanations to illustrate its innovation. All the changes have been highlighted in the manuscript.   

  We hope that these corrections and revisions are satisfactory, and the improved version will be acceptable for publication in Journal of Functional Biomaterials

Thank you for considering our submission, and we look forward to hearing from you.

.

Sang Mo Kwon, PhD

Professor of Department of Physiology, School of Medicine,

Laboratory of Vascular Medicine & Stem Cell Biology

Pusan National University, Yangsan, 626-870, Korea.

Tel: +82 51 510 8070; Fax: +82 51 510 8076; Email: [email protected]

Reviewer 2 Report

The Review paper entitled "Development of Biocompatible 3D Printed Artificial Blood 2 Vessels through Multidimensional Approaches" offers valuable insights into the field of blood vessels regeneration.
However, while it contains a wealth of information, there is an opportunity to streamline the content and enhance its clarity. Specifically, most of the sections appear somewhat verbose and may benefit from being more concise and to the point.
The manuscript in its current version provides an excellent overview of bioprinting techniques, but the focus stays too general while it should be done on blood vessels bioprinting only applications. There is a need for more detailed discussions about how these techniques can be tailored to meet the specific requirements of blood vessel engineering by providing detailed published examples and incorporating case studies or examples of bioprinting applications that specifically pertain to blood vessels. These real-world examples will illustrate the current potential and challenges to discuss then about the future developments needed.

To improve the impact, consider to detail how bioprinting techniques can be customized for blood vessel engineering, including stiffness, resolution, size and vasculature network complexity. Figures of the current published studies to showcase specific techniques and summarize the current possibilities in bioprinting will also be beneficial.
It will enhance the paper's focus and visual appeal, making it an even more valuable resource for the scientific community.

Examples of improvements possible:
- Page 1 line 20 "Integrating these techniques offers the prospect of crafting artificial blood vessels with remarkable precision and functionality.": There is not so many details about the blood vessel functionality maintained by bioprinting.
- Page 1 line 24 "By mimicking the natural complexity of blood vessels,...": More details should be provided to explain about this complexity and what are the current limitations of the published models using bioprinting. How bioprinting can help to overcome them and where are the limits?
- Section 2.1: Please detail some examples of studies using polyglycolic acid and polylactic-co-glycolic acid for blood vessels bioprinting and what were the advantages and limitations?
- Section 2.2 and 2.3: same here, which ECM finally showed the best results for the specific blood vessels bioprinting application?
- Section 2.4: Hydroxyapatite and titanium alloys seem surprising for blood vessel bioprinting application, please detail more the current examples here too.
- Section 2.5 "Establishing a functional network of blood vessels within a construct presents a complex hurdle for oxygen and nutrient exchange and overall construct viability. Selecting a suitable bioink that supports cell growth, maintains structural integrity during printing, and guarantees post-implantation survival is critical": More details are needed about the challenges specifically for blood vessels structures and thus which bioprinting type or which material would be more suitable?
- Section 2.6: this section is interesting but again, examples need to be detailed.
- Section 3.1 "However, challenges include balancing the print speed with cell viability, selecting suitable bioink properties, and developing functional vascular networks within the construct.": Here again, please detail more all these points and provide examples of suitable conditions or limitations.
- Section 3.2 "desired blood vessel structure is created, detailing the dimensions, branching patterns, and architectural features.... The advantages of inkjet bioprinting include its ability to achieve high-resolution printing, gentle handling of living cells, and the potential for automated production [166-169].": Details about the desirable size and resolution for bioprinting are needed.
- Section 3.3 "This technique offers distinct advantages in terms of its high precision, smooth surface finish, and potential to incorporate various biomaterials.": Why is it required for blood vessels application? Which size and resolution?
- Section 3.4 "The strength of SLS lies in its material diversity, which enables the use of a wide range of biomaterials, and its capacity to fabricate intricate geometries.": Is it suitable for blood vessels?
- Section 3.5 "This results in a nanofibrous scaffold that mimics the intricate architecture of natural tissues.": Can this be applicable for blood vessels bioprinting? What is the current size limitation for this technic?
- Section 3.6 "Various assessments have been conducted to evaluate the mechanical properties, structural robustness, and overall functionality of the printed blood vessels.": In summary, what are the known examples suitable for blood vessels in terms of biocompatibility and final characteristics?
- Figure 2: This figure is too general and should be modified to emphasize and summarize the advantages and disadvantages when applied to blood vessels bioprinting in particular.
- Section 4.1 "The prospects of successful organ transplantation are augmented by integrating these engineered constructs, providing newfound hope for patients in dire need of life-saving interventions [48,195-198].": It seems interesting, which materials and printing method were used here?
- Section 4.2 "By inducing specific disease conditions within these constructs, scientists can observe how drugs interact with the diseased blood vessels; thereby, identifying potential therapeutic interventions and elucidating their underlying mechanisms [199-204].": Same here, details and examples of blood vessels applications are needed.
"This deeper understanding is invaluable for uncovering novel targets for therapeutic interventions and allowing the development of drugs that precisely target specific pathways and components within the vasculature.": Here also concerning which disease and which specific target?
- Section 4.3 "scientists can investigate the mechanics of blood vessel behavior by studying factors, such as blood flow dynamics, endothelial cell interactions, and vessel elasticity.": Studies using models to answer these questions were already done and published, but what are the current limitations in the models?
"Researchers can recreate disease-inducing conditions, such as plaque deposition in atherosclerosis, dysregulation of blood pressure in hypertension, or weakening of vessel walls in aneurysms.": Same for this sentence as well, how bioprinting can improve the current models?
Line 488: Can you provide examples of current models using bioprinting for organ-on-a-chip?
- Conclusion section "Collaborations across fields, such as 3D printing, nanotechnology, and biomaterial science can yield hybrid constructs with exceptional properties, ...": For instance? w
What should be developed?

Author Response

(The authors gave the same response as above.)

Reviewer 3 Report

In this review article, Choi and colleagues have described the biofabrication processes used to create vascular and vascularized constructs, focusing on 3D bioprinting. The various techniques used in vascular engineering, including extrusion-, droplet-, and laser-based bioprinting methods are explained in this review article. Additionally, they have described the applications of 3D bioprinted artificial blood vessels. The paper is organized well. However, the following points are strongly suggested to be addressed prior to possible publication:

1.       Figures are an important part of an article. There is only one figure in the article other than the graphical abstract, and this figure is a general demonstration of 3D printing schematics. It is strongly recommended that the authors include a new figure(s) about the topic of the review (e.g., 3D Printed Artificial Blood Vessels) using experimental reports.

2.       It is suggested to enrich the last part of the review about biocompatibility and clinical translation of 3D-printed products using the articles doi.org/10.3390/mi14061099 and doi.org/10.3390/mi13071099

The English is good.

Author Response

(The authors gave the same response as above.)

Round 2

Reviewer 1 Report

The authors have addressed most of the concerns satisfactorily. However, the manuscripts needs some modification to improve the readability and quality. The following points may be considered-

1. Text in Fig. 4 is not legible. Authors may consider rearranging the contents of the figure into separate tables or figures.

2. Authors may consider including ranges of numerical values like resolution, feature size, mechanical properties etc. for different processes and materials, in tabular form or in images.

3. In subpoint 2.4, since authors have mostly discussed about bone and tissue engineering, it would be better if different factors which control vascularization (for example, porosity, pore size, and pore connectivity of metal/ ceramic implants and coating) are considered. Different design strategies which deal with those aspects (doi: 10.1089/ten.tea.2010.0148, 10.3390/bioengineering10060675, 10.1007/s40843-017-9091-1 etc.) may be studied.

Author Response

We wish to submit our second revised manuscript (Review Article, No. jfb-2631892) entitled “Development of Biocompatible 3D Printed Artificial Blood Vessels through Multidimensional Approaches” for your consideration for publication in the Journal of Functional Biomaterials. We express our sincere appreciation to you and the reviewers for your valuable comments and feedback on our previous submissions. We have taken great care in addressing each comment and have undertaken the necessary steps to enhance the quality of our manuscript. In response to the reviewers' comments, we have provided a detailed point-by-point response in the answer sheet accompanying this submission. These revisions have been implemented with the aim of improving the clarity and impact of our study. Changes made to the manuscript have been clearly highlighted for your convenience.     

  We would like to thank you once again for your time and consideration. It is our sincere hope that you find our revisions to be satisfactory, and that our manuscript is deemed suitable for publication in the Journal of Functional Biomaterials.

We appreciate your consideration of our submission and look forward to hearing from you.

Sang Mo Kwon, PhD

Professor of Department of Physiology, School of Medicine,

Laboratory of Vascular Medicine & Stem Cell Biology

Pusan National University, Yangsan, 626-870, Korea.

Tel: +82 51 510 8070; Fax: +82 51 510 8076; Email: [email protected]

Reviewer 2 Report

I would like to thank the authors for their prompt response and the thorough revisions they hae made to the manuscript. I'm pleased to see that the added details and examples of studies have significantly enhanced the quality of the work.

One more comment concerns the Figure 2 (Figure 4 in the revised version). The added details really made the figure more relevant for the blood vessel bioprinting application and provided valuable information. However, I would recommend summarizing and simplifying the text to make it more concise and easier to read. The current density and small size of the text can be challenging for readers to absorb efficiently. Streamlining the content in this figure will enhance its clarity and overall effectiveness in conveying the key points.

Author Response

(The authors gave the same response as above.)
